# Magnetically Driven Biopsy Capsule Robot with Spring Mechanism

**DOI:** 10.3390/mi15020287

**Published:** 2024-02-18

**Authors:** Md Harun Or Rashid, Feng Lin

**Affiliations:** 1School of Mechanical Engineering and Automation, Beihang University, Beijing 100191, China; linfeng@buaa.edu.cn; 2Beijing Advanced Innovation Center for Biomedical Engineering, Beihang University, Beijing 100191, China

**Keywords:** capsule robot, biopsy, active locomotion, electromagnetic actuation system

## Abstract

In recent years, capsule endoscopes (CEs) have appeared as an advanced technology for the diagnosis of gastrointestinal diseases. However, only capturing the images limits the advanced diagnostic procedures and so on in CE’s applications. Herein, considering other extended functions like tissue sampling, a novel wireless biopsy CE has been presented employing active locomotion. Two permanent magnets (PMs) have been placed into the robots, which control the actuation of the capsule robot (CR) and biopsy mechanism by employing an external electromagnetic actuation (EMA) system. A spring has been attached to the biopsy mechanism to retract the biopsy tool after tissue collection. A camera module has also been attached to the front side of the CR to detect the target point and observe the biopsy process on the lesion. A prototype of CR was fabricated with a diameter of 12 mm and a length of 32 mm. A spring mechanism with a biopsy needle was placed inside the CR and sprang out around 5 mm. An in vitro experiment was conducted, which demonstrated the precise control translation (2 mm/s and 3 mm/s in the x and y directions, respectively) and desired extrusion of the biopsy mechanism (~5 mm) for sampling the tissue. A needle-based biopsy capsule robot (NBBCR) has been designed to perform the desired controlled locomotion and biopsy function by external force. This proposed active locomoted untethered NBBCR can be wirelessly controlled to perform extended function precisely, advancing the intestinal CE technique for clinical applications.

## 1. Introduction

The most commonly known diseases of the gastrointestinal (GI) tract are tumors, ulcers, and bleeding, which can cause serious illnesses and internal damage like cancer in the GI tract if they are not properly diagnosed and treated on time. For this diagnosis of GI diseases, endoscopies are commonly used, although they may pose discomfort and certain risks to the patient during anesthesia and operation [1,2]. In 2000, a revolutionary pill-sized swallowable capsule endoscopy (CE) was introduced by an imaging company from Israel and approved in 2001 by the US Food and Drug Administration to examine and capture the picture or video of the diseases via passive movement in GI peristalsis, which helps to diagnose these GI diseases. CE has been broadly used for the medical examination of the GI tract [3].

In recent years, various CEs have made outstanding progress; still, their clinical functions have a lot of limitations at this point. As the demands for CE increases in clinical applications, CE also needs to be advanced with more required functions. Several shortcomings of current CEs compared to conventional invasive endoscopes are biopsy function, drug delivery, anchoring, and weak active locomotion. Nowadays, researchers are performing research to merge CEs into robots with the intention of making an active locomotive robot with extended functions like biopsy, named the capsule robot (CR) [4]. Capsule robots with biopsy functions have become a very important research direction for their effective performance, helping the doctor examine their patients’ conditions [5]. Different types of biopsy tools have been designed to integrate with the CE. Needle-[6,7,8], blade-[9,10,11,12,13,14,15,16], or forcep [17,18,19,20]-type biopsy tools are designed to extract the tissue sample from the GI tract. Since cutting the tissue for sampling or tearing it using forceps may cause wounds or discomfort to the patient, biopsy needles (BNs) have been broadly used in the biopsy techniques of liver, breast, or kidney diseases. The BN technique has been successfully implemented to merge with CE. For example, Son et al. designed a biopsy CR with a fine needle and performed their experiment in vitro, which resulted in their successful tissue collection [7]. Meanwhile, Ye et al. performed an ex vivo experiment using the BN technique and collected their sample tissue successfully. The average tissue sample length was about 1 mm [8]. Compared to BNs, biopsy blades or forceps can collect the sample more effectively, but these can only collect samples from the surface of the GI and cannot penetrate inside. Also, these techniques can cause feelings of discomfort in the patient and hurt the wall of the GI tract. For example, Le et al. introduced a biopsy CR with forceps, which needed a 1.32 N force to conduct the experiment [20], which is deemed slightly dangerous for the GI wall.

Nowadays, the common problem faced by the available commercial CEs is active locomotion, since it passes randomly and passively through the GI tract. It also makes it difficult for the CRs to perform the biopsy in the desired location [8]. To address this shortcoming, researchers have studied it, and two types of active locomotion of CE have been found so far in terms of the actuation method, which are the internal and external actuation systems. The common types of internal actuators are motor-driven legs [21,22], spiral [23], or bionic worm mechanisms [24], whose most important mechanism is the motor. This is why these actuators are less effective, since motors consume more energy. In contrast, external actuators consume less energy and move remotely. In order to achieve this, one or more PMs can be installed into the CE so that it can be driven by using an EMA system [25] or PM [26,27,28]. It is also convenient to control the external magnetic system while reducing energy consumption. This EMA system also solves the triggering problem of biopsy function by installing internal PMs. Several types of triggering devices have been used so far to actuate the biopsy mechanisms, e.g., motors [7,19], heat-sensitive parts [9,13,17], or permanent magnets (PMs) [6,8,10,11,12,14,18,20]. Among them, motors can drive the biopsy tool quickly and can be controlled simply, but they consume more energy. The heat-sensitive parts are mainly made from shape memory and paraffin wax, which draw energy from the inner battery of the CR. It can also burn the GI tract due to its high temperature and can be performed only once due to its characteristics. Compared to these actuators, PMs are the perfect actuation system. The electromagnetic actuation (EMA) system is used for this PM actuation system. It does not consume energy from the internal battery and can be controlled remotely, consequently saving the internal space and making it easier to install and move the CR in the simulated gastric environment.

In this paper, a novel needle-based biopsy capsule robot (NBBCR) has been proposed for tissue collection, where a BN is merged with the available commercial CE. A spring mechanism is designed to control the triggering of the biopsy mechanism. The external magnetic field (EMF) is used to push the biopsy punch toward the GI wall, and a spring mechanism is designed to retract the punch needle. The BN stays inside the CR so that the needle does not perforate any undesired area on the GI wall during locomotion and pushes outside when the CR reaches the desired location to collect tissue samples using the rotational motion of the EMF. After the tissue collection, the EMF is turned off, and the spring pushes back the punch needle inside. A camera is also used to see inside the GI tract. This proposed CR is remotely controlled and enabled for scanning, active locomotion, and tissue collection to diagnose intestinal diseases.

The organization of this paper from the next section onwards is as follows: The design and working principles are discussed in Section 2. The control strategy is presented in Section 3. Section 4 represents the experiments and results. The discussion and conclusions are presented in Section 5 and Section 6, respectively.

## 2. The Development of the Needle-Based Biopsy Capsule Robot

### 2.1. Design Structure of the Capsule Robot

A vitamin pill-sized NBBCR was designed that is suitable for oral administration. The designed NBBCR is 12 mm in diameter and 32.2 mm in length (see Table 1), which was designed based on the available commercial CE [4]. It is composed of a camera module with a camera, a fine needle, a spring, two PMs, and signal transfer modules (see Figure 1). PM 1 is placed on the front side of the CR and is used to control the locomotion and rotation of the CR and help the CR stick to the target position. Based on the properties of the magnets, neodymium–iron–boron (NdFeB)-type permanent magnets were used. This type of magnet has a higher energy yield, which makes it suitable for the compact design of innovative applications and lowers the production cost. Based on the designed requirement and availability on the market, 2 magnets with a 10 mm outer diameter, a 5 mm inner diameter, and a 3 mm thickness were selected for PM 1. For PM 2, a magnet with a 9.5 mm outer diameter, a 6 mm inner diameter, and a 3.5 mm thickness was chosen, which was used to control the biopsy mechanism to help spring it out. The biopsy mechanisms and camera module are stowed at the front part of the CR so that the CR can easily visualize the GI tract during locomotion and help to precisely control movement and tissue collection. The biopsy mechanisms consist of a compression spring and a fine needle, which are pushed by PM 2. The needle is 1.2 mm in diameter and 23 mm in length, and it will be pushed outside around 4~5 mm to punch the GI’s wall when the external magnetic force (EMF) is applied. After the tissue collection, the EMF is turned off, and then the compression spring pushes the needle back inside the CR. The camera is placed next to the needle so that the biopsy needle can be triggered at the desired location of the selected lesion. The signal transfer module is placed on the rear side of the CR.

### 2.2. Working Principles of the Capsule Robot

The proposed biopsy capsule robot is designed to perform tasks under an external electromagnetic force. The EMA system was used due to its significance in producing a controllable uniform and gradient magnetic field, which enables us to generate magnetic torque and force acting on the permanent magnet, steering the capsule and pushing it along the planned path. By driving the internal permanent magnet using the external EMA system, the robot can achieve active translational and rotational motion and can also perform the biopsy operation for further diagnosis. This system also enables us to reduce the space consumption of the battery or other internal power supply system for the locomotion of CR, so the spatial potentials can be used for other purposes. As the NBBCR is designed to sample the tissue from the tumor-type lesion of the GI tract (large/small intestine), the robot will move through it. As shown in Figure 2a, CR collects the tissue sample when it reaches the lesion. The process of tissue collection can be understood by the cycles of working principles of NBBCR, which are in 4 stages (see Figure 2b).

The CR will first move through the GI tract through translation motion control by gradient EMF acting on PM 1. PM 1 will act as a driving magnet during this locomotion. At that time, the EMF on PM 2 will be smaller than the force on PM 1, so the robot will be driven according to PM 1. Therefore, the needle will not come out during this active locomotion. After that, when the NBBCR reaches the target point, a magnetic torque will be applied to PM 1 to rotate the CR to place the needle toward the lesion and stick the robot with the tumor, making it unable to slip in the 2nd stage. When the needle is pointing out toward the lesion, a uniform magnetic force will be applied to PM 2, which is larger than the spring force. Since the opposite directional force is applied to PM 1 and PM 2, locomotion will not happen at this stage. Therefore, the needle will spring out and punch the target location for tissue collection in the 3rd stage. For this NBBCR, the needle is designed to come out around ~5 mm, which is the appropriate penetration depth for tissue collection by needle mentioned by Son et al. [7]. Again, a rotational force will be applied to PM 1 to tear off the tissue. After the tissue collection, the EMF will be turned off or reduced to less than spring force on the PM 2 so that the spring can push back the needle to retract it in the last step. Therefore, the needle will not damage any other part of the GI’s wall during translational motion.

## 3. Control Strategy of the Capsule Robot

### 3.1. The Electromagnetic Actuation System

The control strategy of CR movement is very important to achieve a successful diagnosis of the GI tract. The conventional endoscopic capsule has passive locomotion; thus, its movement is controlled by the peristaltic movement of the digestive organs. This passive motion leads to unsuccessful diagnosis and therapeutic motion in small hollow organs and unreliability in digestive organs such as the stomach and colon. Therefore, the active locomotion of CE must be implemented. In this study, an EMA system has been utilized to achieve active locomotion with wireless control for the endoscopic capsule. This control actuation system was introduced by Song et al. to study the magnetic levitation control of CE [26].

It is in Figure 3 that the EMA system consisted of 1 hollow cylindrical coil and 6 hollow square coils with almost the same specifications and configurations. The configurations are shown in Table 2. The square coils were named orientation control electromagnetic coils (OCECs)s and the cylindrical coils were named magnetic levitation control electromagnetic coils (MLCECs). For this paper, the OCEC coils were used for the translational motion and rotation of the CR in the working space, and the MLCEC coil was used to control the spring function. These six OCEC coils are aligned with the origin at the same center coordinate, and their distances are equally distributed from the origin, maintaining adequate physical workspace for the human body with a smaller overall device size. MLCEC, a single electromagnetic coil, is placed on a 3D moving platform (see Figure 3) at an angle of about $45^{\circ }$. When necessary, MLCECs can extend from the top of the OCECs into the capsule robot’s workspace for the motion control of the spring. To control the motion of the CR, the coils were designed accordingly to produce the required magnetic field.

To analyze the distribution of the magnetic field, a finite element simulation method was conducted using COMSOL Multiphysics 6.0 software. The copper wire has been used for the coils, whose arrangements are shown in Figure 3a, and the specifications are mentioned in Table 2. The magnetic field distribution in the workspace of OCEC coils is shown in Figure 4a, where the input currents of the coils in the simulation were 2 A equally. The distance between each pair of coils was 205 mm, which was the minimum spacing between two coils, as the lesser spacing generated a larger magnetic flux density [29] that is necessary for this project.

The simulation result shows a strong uniform magnetic field distribution with a maximum of around 30 mT when 2 A equal current is applied to each of the 6 coils. As the same input current is applied, a neutral zone is created at the center, at an equal distance from all coils. Next, when the different input current (less in coil 2) is applied, the neutral position changes from the center towards coil 2’s direction showing by the arrow (see Figure 4b). Similarly, when less current flows in coil 2 or more current flows in coil 1, the neutral position is changed from the center toward coil 2. By applying this method, the forward and backward movements of the robot can be controlled precisely.

### 3.2. Magnetic Field Analysis

The magnetic field of an EMA system is calculated by the current (*I*) flowing through the coils, where the current is defined in terms of the voltage ($V_{i})$, resistance ($R_{i}),$ and inductance ($L_{i})$ of the coils. So, the current through an inductor can be obtained using a differential equation:(1)$$\frac{dI}{dt}=-\frac{R_{i}I}{L_{i}}+\frac{V_{i}}{L_{i}}$$

In this study, fixed electromagnetic coils were utilized to produce a dynamic magnetic field. In a control volume, the superposition principle can be used to determine the magnetic field at any position in space. Hence, the generated magnetic field of all coils can be obtained separately and then added them together. To determine the generated magnetic field of each OCEC and MLCEC, Biot–Savart law should be applied:(2)$$B=\frac{\mu_{0}N_{t}I}{4\pi }\oint_{s}^{\;}\frac{d\mathrm{l’}\times a_{R}}{R^{2}}$$
where *B* represents the magnetic field at the place of the capsule robot, which can be written as *B*$={\left(B_{x}\;B_{y}\;B_{z}\right)}^{T}$ in cartesian coordinates; $\mu_{0}\;(\mu_{0}=4\pi \times 10^{-7}\;N/A^{2})$ is the vacuum permeability; $N_{t}$ is the number of turns of a specific coil; *I* represents the input current of the coil; dl’ is the infinitesimal line segment of the integral; $a_{R}$ represents the unit vector from the product component of the line segment to the target position; and *R* is the distance from the product component of the line segment to the target point. For every square coil of OCEC, the contour integral has to be converted into an integral of four sides [30]. As for the MLCEC, Equation (2) can be directly used to calculate the magnetic field distribution. The distribution of the magnetic field along the *x*-axis for each square coil can be evaluated in the Cartesian coordinate system by the equation formulated below:(3)$$r=\sqrt{{\left(x-c\right)}^{2}+{\left(y-y_{i}\right)}^{2}+{\left(z-z_{i}\right)}^{2}}$$
(4)$$B_{x}=\frac{\mu_{0}NI}{4\pi }\sum_{i=1}^{2}\sum_{j=1}^{2}{\left(-1\right)}^{i+j}\times \frac{\left(z-z_{i}\right)\left(y-y_{i}\right)}{r}\times \left[\frac{1}{{\left(x-c\right)}^{2}+{\left(y-y_{i}\right)}^{2}}+\frac{1}{\left({\left(x-c\right)}^{2}+\right){\left(z-z_{i}\right)}^{2}}\right]\;\;\;\;\;\;\;\;\;\;\;\;\;\;\;$$
(5)$$B_{y}=\frac{\mu_{0}NI}{4\pi }\sum_{i=1}^{2}\sum_{j=1}^{2}{\left(-1\right)}^{i+j+1}\times \frac{\left(z-z_{i}\right)\left(x-c\right)}{r}\times \left[\frac{1}{{\left(x-c\right)}^{2}+{\left(y-y_{i}\right)}^{2}}\right]$$
(6)$$B_{z}=\frac{\mu_{0}NI}{4\pi }\sum_{i=1}^{2}\sum_{j=1}^{2}{\left(-1\right)}^{i+j+1}\times \frac{\left(y-y_{i}\right)\left(x-c\right)}{r}\times \left[\frac{1}{{\left(x-c\right)}^{2}+{\left(z-z_{i}\right)}^{2}}\right]$$
$$z\;\epsilon \;\left[a,-a\right],\;y\;\epsilon \;\left[a,-a\right]$$
where $B_{x},\;B_{y},\;{\mathrm{and}\;B}_{z}$ are the magnetic field intensity components of the magnetic field generated by a square coil comprising an OCEC along the coordinate axis, while *c* is half of the effective side length of the square coil, and x  is the coordinate of the coil center position on the *x*-axis. The total magnetic field distribution generated by the total OCEC system can be obtained by the superposition principle.

### 3.3. Magnetic Force Analysis

The strength of a magnetic field is affected by the force and torque of the magnetic field, and its strength size depends on its magnetic field intensity or magnetic field gradient. When the magnetic field is generated by OCEC and MLCEC coils and satisfy the superposition principle, the total magnetic flux density, *B*, is superimposed by multiple independent magnetic flux density. Then the total force, *F*, and torque, *T*, on the magnetic field are satisfied by the following equations, respectively:(7)$$F=V\left(M.\nabla \right)B=\left[\begin{array}{ccc} \frac{\partial B_{x}}{\partial x} & \frac{\partial B_{x}}{\partial y} & \frac{\partial B_{x}}{\partial z}\\ \frac{\partial B_{y}}{\partial x} & \frac{\partial B_{y}}{\partial y} & \frac{\partial B_{y}}{\partial z}\\ \frac{\partial B_{z}}{\partial x} & \frac{\partial B_{z}}{\partial y} & \frac{\partial B_{z}}{\partial z} \end{array}\right]VM\;\;\;\;\;\;\;$$
(8)$$T=VM\times B=\left[\begin{array}{ccc} e_{x} & e_{y} & e_{z}\\ VM_{x} & VM_{y} & VM_{z}\\ B_{x} & B_{y} & B_{z} \end{array}\right]$$
where *B* is the total magnetic flux density of the capsule robot at the position *x*, *y*, or *z* and is in the Cartesian coordinate system; the value can be $B={\left[B_{x},\;B_{y},\;B_{z}\right]}^{T}$. The volume of the cylindrical permanent magnet is represented by *V*. The magnetization per unit volume, *M*, is represented by $M={\left[M_{x},\;M_{y},\;M_{z}\right]}^{T},$ and ${\left[e_{x},\;e_{y},\;e_{z}\right]}^{T}$ represents a unit matrix. ∇ is a gradient operator, and “.” represents the dot product. Considering there is no current flowing in the working space and the quasi-static magnetic field is satisfied, the following equations can be achieved:(9)$$F=\left[\begin{array}{ccc} \frac{\partial B_{x}}{\partial x} & \frac{\partial B_{x}}{\partial y} & \frac{\partial B_{x}}{\partial z}\\ \frac{\partial B_{y}}{\partial x} & \frac{\partial B_{y}}{\partial y} & \frac{\partial B_{y}}{\partial z}\\ \frac{\partial B_{z}}{\partial x} & \frac{\partial B_{z}}{\partial y} & -\frac{\partial B_{x}}{\partial x}-\frac{\partial B_{y}}{\partial y} \end{array}\right]\left[\begin{array}{c} VM_{x}\\ 0\\ VM_{z} \end{array}\right]$$
(10)$$T=\left[\begin{array}{ccc} 0 & B_{z} & {-B}_{y}\\ -B_{z} & 0 & B_{x}\\ B_{y} & B_{x} & 0 \end{array}\right]\left[\begin{array}{c} VM_{x}\\ 0\\ VM_{z} \end{array}\right]$$

When the CR reaches the target position by translational motion, it requires a rotational motion generated by the OCEC, positioning the CR at the same center line as the MLCEC. At that time, the CR creates a tilt angle, φ, along the combined OCEC magnetic field’s direction. The tilt angle and force situations are shown in Figure 5, where the CR transformed the 3D to 2D XZ plane about Y. During stable movement and controlled angles, several forces and torques are generated at the center of mass. Force, $F_{x1}$, by the MLCEC in the x-direction; force, $F_{x2}$, by the OCEC in the x-direction; and force, $F_{z1}$, by the MLCEC in the z-direction. $F_{z2}$, force by the OCEC in the z-direction, $T_{y1}$, torque by MLCEC,$\;T_{y2}$, torque by OCEC, buoyancy force, $F_{\rho },$ and CR’s weight, *mg*, are acted on. The magnetization vector, *M* $\left(M={\left[\begin{array}{ccc} M_{x} & M_{y} & M_{z} \end{array}\right]}^{T}\right),$ changes to the $M={\left[\begin{array}{ccc} M_{x} & 0 & M_{z} \end{array}\right]}^{T}$ form due to static motion. So, the tilt angle can be written as follows:(11)$$tan\varphi =\frac{sin\varphi }{cos\varphi }=\frac{M_{x}}{M_{z}}$$

If it is assumed that the OCEC system generates the magnetic field, whose direction is located on the ZX plane shown in Figure 5, and the capsule has stable motion, then the following dynamic relations can be formulated:(12)$$m\ddot{x}=F_{x,1}-F_{x,2}$$
(13)$$m\ddot{z}=F_{z,1}-F_{z,2}-mg+F_{\rho }$$
(14)$$J\ddot{\varphi }=T_{y,1}-T_{y,2}$$
where *J* is the polar moment of inertia of the CR, and m represents the mass of the CR. As it has been assumed that the robot has stable movement, Equations (12)–(14) can thereby be considered almost static [31]. Therefore, a relatively uniform magnetic field can be considered generated by the OCECs, there will be balanced magnetic torque, and the properties of the magnetic field generated by the OCECs can be ignored. The equation can be formed as:
(15)$$tan\varphi =\frac{B_{x,1}-B_{x,2}}{B_{z,1}-B_{z,2}}$$
where $B_{x,1}$ and $B_{x,2}$ represent the magnetic flux generated by the MLCECs and OCECs, respectively, in the *x* component, and $B_{z,1}$ and $B_{z,2}$ represent the magnetic flux generated by the MLCECs and OCECs, respectively, in the z component with the function of the (x, z) position of CR, current $I_{1}$ flowing into the MLCECs, and current $I_{2}$ flowing into the OCEC. The balanced magnetic field components of the MLCEC and OCEC systems are provided by the tilt angle. The orientation of the tilt angle is determined by the direction of the OCEC’s magnetic field based on previous assumptions, while the value of the tilt angle is associated with the magnetic field components produced by the OCECs and MLCECs on the ZX plane.

### 3.4. Cutting Force and Sampling

The tissue collection process of the biopsy module is composed of two steps. In the first step, the robot will push the biopsy tool against the GI wall into the lesion, and the biopsy tool will penetrate the tissue. It will create a cutting imprint on the lesion’s surface when the combined shear force and spring force are applied. In the second step, the capsule robot will provide a rotational motion to tear off the tissue and samples. It is seen from Figure 6 that a combination of two forces (shear force, $F_{n},$ and spring force, $F_{k}$) are required to push the biopsy tool into the lesion. When the CR reaches the target region, it will rotate pointing to the lesion and stick the robot with the GI wall by applying force to PM 1. Then a uniform magnetic force, $F_{TC}$, to cut the tissue, will be applied to PM 2 that will push the needle out to insert into the GI wall. After piercing the needle into the tissue, the CR will rotate about the *y*-axis to create a destructive shear stress so that the needle can tear off the tissue using force on PM 1. The force to cut the tissue can be expressed as $F_{TC},$ and the required force will be as follows:(16)$$F_{TC}>F_{K}+F_{n}$$
where $F_{k}$ is the spring force and $F_{n}$ is the required punching force. The magnetic force, $F_{TC},$ has to be greater than the summation of spring force, $F_{k},$ and punching force, $F_{n}$. The spring force, $F_{k}$, can be calculated as follows:(17)$$F_{k}=kx$$
where *k* is the spring stiffness and *x* is the distance. To calculate the spring force, it mostly depends on the spring stiffness of the used spring, as the distance, *x*, to push the needle is fixed. The appropriate penetration distance of the needle into the lesion is around 5 mm [7]. For this project, a cylindrical compression spring has been used, which has a 0.2 mm wire diameter and ~70 N/m stiffness. The 17 mm length of the spring has been placed inside the robot, and only around 5 mm will be pushed.

This spring required relatively little force to push but has enough strength to hold back the biopsy mechanism during translational motion and retract their original position. So, the total spring force is estimated to be more than 0.35 N. For the required punching force, $F_{n}$, it depends on the type of needle. For the biopsy function, a needle from 19 G (gauge) to 25 G (gauge) that is suitable for this process will be used [8]. The sharpness, the inner and outer diameter of the needle to determine the cut section area, $A_{c}$, and the destructive shear stress, $\tau_{d}$, are required to determine the punching force. The formulas are as follows:(18)$$F_{n}\geq \tau_{d}{\;\times \;A}_{c}$$
(19)$$A_{c}=s\;\times \;\pi \;\times \;d_{punch}$$
where *s* is the sharpness and $d_{punch}$ is the diameter of the tip of the biopsy punch. In this study, a punch tool with outer and inner diameters of 1.5 and 1.2 mm, respectively, and a sharp tool tip of 0.03 mm were used. The suitable destructive shear stress for biopsy needles is proposed by Hoang et al., which is 1.2 MPa [6]. Therefore, the required shear force, $F_{n}$, is estimated to be at least 0.17 N ($d_{punch}$ = 1.5 mm). Finally, the total force needed to cut the tissue, $F_{TC}$, can be estimated, which has to be ≥0.52 N.

Now the rotation motion of the capsule body is used to tear off the tissue. The required magnetic torque can be obtained from the following equation:(20)$$\left\|T\right\|=F_{TC}\;\times \;d$$
where *d* = 14 mm is the distance from the center of PM 1 to the tip of a 5 mm extruded punch tool. Accordingly, the capsule must be able to create a minimum of 0.0073 Nm of physical torque.

## 4. Experiments and Results

### 4.1. Experimental Process

For the experimental setup, an EMA system was developed, as illustrated in Section 3.1, with some modifications that were used by Song [26]. The MLCEC was placed at an angle of around $45^{{}^{\circ}},$ as shown in Figure 7a, to apply the uniform magnetic force to PM 2. The distance between each pair of coils was 130 mm for x and y directional motion. To perform the experiment, the proposed NBBCR was fabricated and all parts assembled. The outer cover and magnet’s cases were designed based on the dimensions demonstrated in Table 1 and printed by a high-precision 3D printer using photosensitive resin materials. The magnets (NdFeB), compression spring, and needle were bought from the online market with the required specifications, and all the parts were assembled, as shown in Figure 7b. A smooth surface dish with a diameter and height of around 80 mm and 10 mm, respectively, used to conduct the experiment, referring to Figure 7c.

To control the current and electromagnetic field, two digital DC power suppliers (Maisheng High Precision Adjustable Digital DC Laboratory Power Supply) were used, which could provide a maximum 100 V voltage and 20 A current. The key feature of this DC power supply is the automatic conversion of constant current or constant voltage (changing one automatically changes the other), which can be easily controlled by rotating a knob. It is also protected from overvoltage, overcurrent, or/and overtemperature. The current into the coils was controlled using the knob of these power suppliers manually during the experiment. The current was input as a requirement (see Table 3) to drive the robot translationally and rotationally and to spring out the needle.

### 4.2. Translational Motion and Control Locomotion

A set of in vitro experiments was conducted to analyze the translational motion and efficiently control active locomotion. Manual control operation was implemented by controlling the digital power suppliers and inputting control current through the coils. As the model of NBBCR was 3D printed and its surface roughness was difficult to measure, resulting in an undetermined frictional coefficient, the sufficient driving motion of the robot was determined by slowly increasing the gradient magnetic field by manually increasing the current through the coils. To determine the translational motion in each direction (x and y), two power suppliers were used for each pair of coils, and different currents were input. The experiments were conducted on a smooth surface dish on which NBBCR was placed, and the dish was placed at the center of the EMA system.

To achieve the x-directional motion, coil 1 and coil 2 were connected with two powers, and current was increased slowly from 0 A to 0.5 A and fixed for coil 2, and 0 A to 2.5 A for coil 1. When the current reached 2.5 A in coil 1, CR received the first motion, and then the current in coil 1 was also slowly reduced as CR was moving near the coil. The results are shown in Figure 8a in chronological order for x-directional translation. At 2 s, the capsule obtained its first motion and changes position around 4 mm until 4 s. From 4 s to 7 s, they moved about 4 mm after some rest. After 7 s, the capsule translated around 20 mm. For y-directional motion, a similar control strategy of the current was applied for coils 3 and 4. The results for y-directional translation are shown in Figure 8b in chronological order. The capsule obtained its first motion and changed position around 3 mm in 1 s. Then, at 5 s, the robot moved around 15 mm after resting. After 5 s, the capsule further translated around 25 mm within a half-second. In the first stage, the CR received its first motion and slightly translated its position.

The graph shows the variation of current control through the coils. In one coil, the current was constant (blue line) while in other, it was variated manually as required (orange line). In the first stage, the current was reduced rapidly after the first motion of the robot to control its translational motion, then increased slowly in the second stage. After starting the movement of the robot in the second stage, the current was again slowly reduced for control motion. The first and second stages of translation received appropriate control motion around 2 mm/s in the x-direction and 3 mm/s in the y-direction and changed the position further. The third stage showed comparatively fast motion.

A forward and backward control of the capsule’s active locomotion was determined by conducting the experiment in the y direction. The capsule was placed at the center of the EMA system and slowly changed the gradient magnetic field by changing the current into coils 3 and 4. The exact idea is shown in Figure 4b,c, which changes the position of the neutral zone inside the magnetic field. For this experiment, first the current was slowly increased to 2 A for both coils. As the current was flowing in the same direction, the robot did not obtain any motion. Then the current was very slowly decreased for coil 3, while it was increased for coil 4, which is clearly shown using the graph in Figure 9, which was plotted using experimental data. Consequently, the CR received a control active locomotion. The current was decreased to 0.5 A for coil 3 and increased to 3.5 A for coil 4. Later on, the current in coil 4 was decreased, and coil 3 was raised to drive the robot in the reverse direction. It was reversing until the current in coil 4 reached 3.5 A and coil 3 reached 0.5 A, then the current was turned off. The current was decreasing and increasing manually but very slowly to obtain the precise motion. The results are shown in Figure 9 in chronological order, which shows the control locomotion of the NBBCR. In forward motion, the robot traveled around 8 mm in the first 20 s (from 5 s to 25 s) with an average velocity of 0.4 mm/s. On the other hand, it is seen that the robot backward traveled by around 4 mm in the last 20 s. Since the current was controlled manually, the capsule did not undergo uniform motion. Sometimes it was rested, then again went through the locomotion. During backward locomotion, the capsule was rested until the reversing force (current in coil 3) increased. While the current in coil 3 flew more than 2 A, the NBBCR was moving in the reverse direction.

### 4.3. Rotating the Capsule and Performing Biopsy Functions

The designed NBBCR has the biopsy mechanism on its front side and in a longitudinal direction, so it is necessary to rotate the robot to point the needle towards the lesion (see Figure 10). Rotation of the CR is also required to stick the CR to the tumor-type lesion and tear off the tissue after pining the needle into the lesion. The robot is also required to rotate to position itself in the same centerline of the MLCEC for uniform magnetic force. The experiment was conducted using coils 5 and coil 6, connecting them together and creating the same directional magnetic field using one DC power supply. The desired rotational angle was between $30^{{}^{\circ}}\mathrm{and}\;45^{{}^{\circ}}$ which could appropriately attach the head of the CR with the lesion. The NBBCR was placed on the dish, and ultraviolet clay was used as an example of a lesion. The current was slowly increasing from 0 A to 1.5 A. The results from Figure 10 show that the angle increases with the increasing magnetic field. As the angle increased, the robot stuck its head into the lesion and pointed the biopsy needle toward it.

To experiment with the biopsy mechanism, two types of experiments were implemented. One experiment was covering NBBCR’s head with ultraviolet clay, and the other was without the cover to show the exact movement of the needle (see Figure 11). To conduct this experiment, the MLCEC was positioned at around $\leq 30^{{}^{\circ}}$. For the first experiment, without covering the head, the robot was placed on ultraviolet clay to make it high from the surface of the dish so that it did not need to face any barriers when it sprung out. The capsule was positioned at the same angle with the MLCEC so that it received uniform magnetic force. Then the uniform force was applied by introducing current through the coil. The current was rising from 0 A to 4 A. Since the north poles of PM 1 and PM 2 were placed side by side inside the robot and the south poles were on opposite sides, the two magnets always retract each other. As PM 1 was designed to drive the whole capsule robot, it was fixed with the robot, and PM 2 was free to move to achieve the biopsy function. So, when uniform magnetic force was applied by coil 7, the whole capsule was attracted by the coil due to the force on PM 1. In contrast, PM 2 was pushed forward as it received retraction force from the coil.

As the magnetic force increased with increasing current in the coil, the retraction force of PM 2 also increased, resulting in the needle bursting out of the robot, as seen in Figure 11a. The current flow through coil 7 is shown in Figure 11b, which is plotted using experimental data. For this experiment, when the current reached 4 A, the length of the needle came out around 5 mm, which met the requirement mentioned by Son et al. [7]. Then the power was slowly reduced and turned off, resulting in the needle being pushed back inside the robot by spring force.

In the next experiment, the robot’s head was covered by ultraviolet clay, which was around 4 mm thick, and the same procedures were applied while the input current was increasing until 5 A. The results in Figure 11 show that PM 1 can produce enough force to overcome the spring force. When the current reached 5 A, the length of the needle came out through the clay at around 1 mm, indicating the feasibility of tissue extraction using NBBCR. A stronger magnetic force was required to overcome the cutting force and pierce the clay. Then the power was slowly reduced and turned off, resulting in the needle being pushed back inside the robot by spring force.

## 5. Discussion

In this study, a biopsy-function-compatible NBBCR has been proposed for future clinical applications. A needle-based biopsy mechanism has been designed inside the robot for tissue collection from the designated GI tract (large/small intestine) for further analysis of diseases. The biopsy tool has been placed in such a position and held back by a spring mechanism so that it does not unintentionally come out and scratch or pierce the intestine’s wall. The key advantage of the proposed CR is the active control strategy using the external EMA system for both locomotion and biopsy application, implementing needle and spring mechanisms. Therefore, the robot does not require any internal power system, resulting in enough free space for other functions. Another key advantage is the strategic placement of the biopsy tool, which is on the front side of the robot, where the camera module has to be placed in a longitudinal direction. So, the robot can easily detect the defect area during active locomotion into the GI tract and easily position the robot in the target position, unlike positioning the biopsy tool in the middle of the robot [8]. The new design can show the whole biopsy procedure to the operator, including the needle sprung out, tissue extraction process, and retrieval of it. Therefore, the proposed design can improve the accuracy of performing biopsies on the target lesion.

The experimental results conducted in vitro showed the appropriate control motion for translational locomotion and rotation, and proper control of biopsy function. The average translational motion in the x and y directions showed an average of 2 mm/s and 3 mm/s, respectively, in the first and second stages after obtaining the first motion, and the forward and reverse control motion showed an average of 0.4 mm/s. The rotation also obtained a slow increase in angle until around $\sim 40^{{}^{\circ}}$, which was enough for the proper positioning of the robot for this study. The key feature, i.e., the biopsy mechanism shows good accuracy in controlling the needle springing out (~5 mm) and retraction, which comply with researcher Son and his team [7].

The overall performance of the proposed design indicates the feasibility of tissue extraction using the NBBCR, though there are still issues where improvement might result in further improved performance of the CR. The first problem is unstable motion control in the third stage when the CR gets closer to the coil. Since the current was controlled manually, am image-based closed-loop control algorithm with a PID controller can be used for motion control by providing real-time feedback. The next problem was the surface friction of PM 2 when force was applied to it. As the magnet’s case and outer shell of the robot were 3D printed, they got stuck due to their surface roughness. When more force is applied to overcome the frictional force, it creates vibrations in the whole robot. Another improvement can be made by replacing the needle with a thinner diameter so that it can easily pierce the tissue.

The proposed design can facilitate endoscopy and biopsy techniques, but it still has limitations for application. The presented NBBCR is designed to perform tasks in the large/small intestine, and the biopsy function can be performed on one lesion several times. If the biopsy is performed on a different lesion, the collected sample tissue will entangle due to the limited space and will lead to an inaccurate diagnosis. In the future, further experiments will be conducted in ex vivo and in vivo using the presented CR to validate the proposed design. The environmental impact on the intestine, which affects the control performance, will also be addressed in the ex vivo and in vivo tests. Other features (camera and signal transfer module) will also be assembled with biopsy function, and an improved control technique and feedback system will be implemented with the EMA system to improve the control accuracy and tract the robot into the GI tract.

## 6. Conclusions

In this paper, a magnetically controlled capsule robot has been presented with a biopsy function for collecting sample tissue from a designated lesion for further analysis. The robot is controlled by an external EMA system for active locomotion and the biopsy application. A 3D-printed model of the robot was fabricated and assembled, and an in vitro experiment was conducted using the model to evaluate the active motion control and frontal biopsy function of the robot. The average control motion in translation and rotation and the spring retractable base needle technique for biopsy application indicate the perfection of utilizing the NBBCR for tissue extraction. Regardless of the performance in vitro, several issues will be further addressed in the future in ex vivo and in vivo experiments for further evaluations of the proposed robot.

## Figures and Tables

**Figure 1 micromachines-15-00287-f001:**
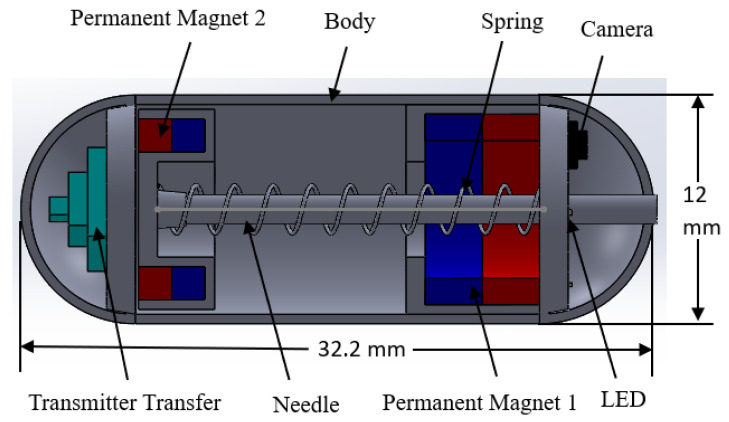
Different parts of the capsule robot.

**Figure 2 micromachines-15-00287-f002:**
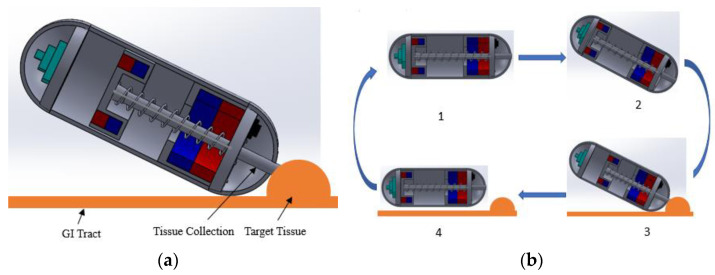
Working principles: (**a**) tissue collection; (**b**) collection procedures.

**Figure 3 micromachines-15-00287-f003:**
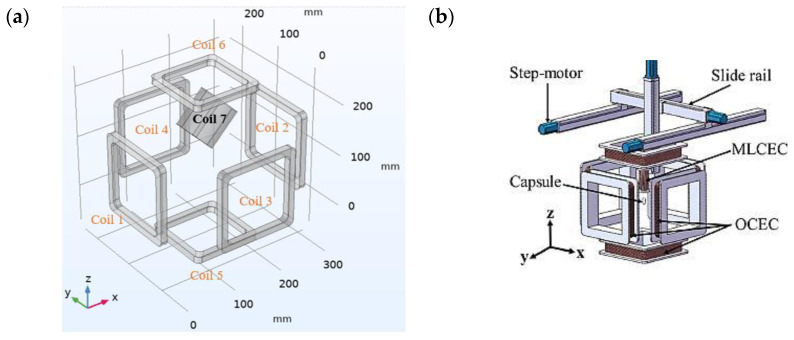
Schematic view of the EMA system: (**a**) coil arrangement; (**b**) coil on actual platform.

**Figure 4 micromachines-15-00287-f004:**
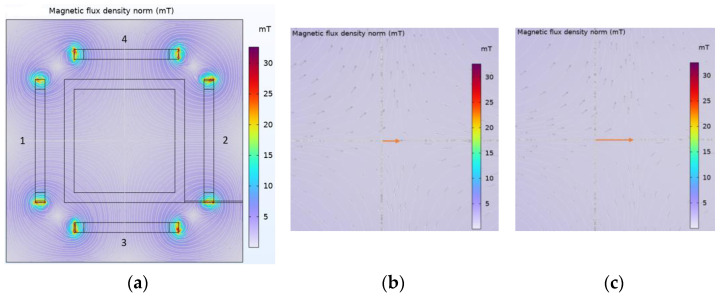
The magnetic field simulation. (**a**) Distribution on the YZ plane, and the distance between the fixed coil pairs is 205 mm. (**b**,**c**) Center moving with the input current (2 A, 1 A, 2 A, and 2 A for coils 1, 2, 3, and 4, respectively, for (**b**) and 2 A, 0.5 A, 2 A, and 2 A for coils 1, 2, 3, and 4, respectively, for (**c**)).

**Figure 5 micromachines-15-00287-f005:**
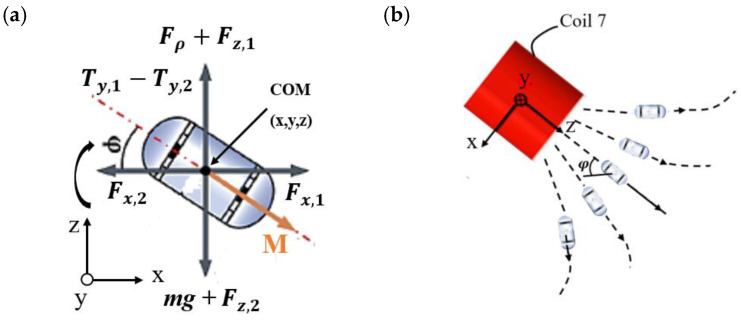
D motion process and force analysis. (**a**) Free body diagram of the capsule robot with the applied forces and torques. M is denoting the magnetization vector. (**b**) The motion state of the capsule robot.

**Figure 6 micromachines-15-00287-f006:**
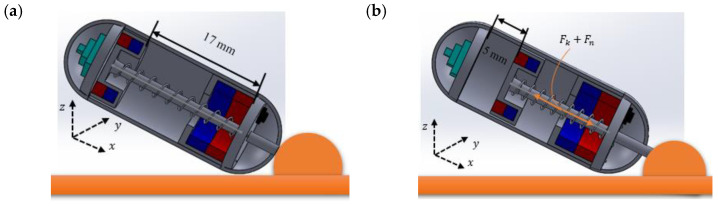
(**a**,**b**) Force analysis of the biopsy module for tissue collection.

**Figure 7 micromachines-15-00287-f007:**
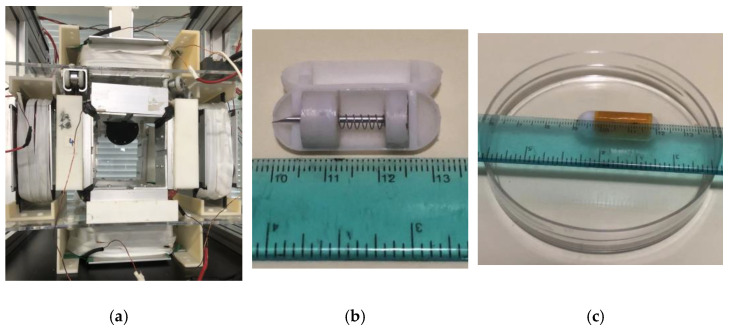
Experimental procedures. (**a**) The EMA system. (**b**) The 3D-printed capsule robot. (**c**) A smooth surface dish.

**Figure 8 micromachines-15-00287-f008:**
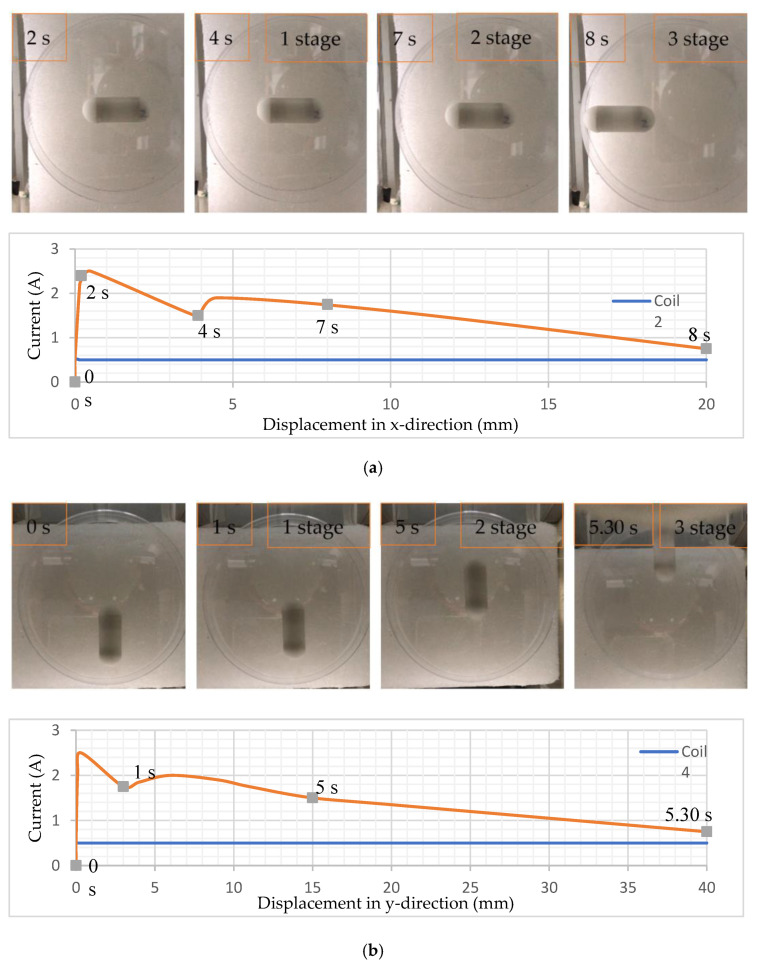
Translational motion and current control graph: (**a**) x-direction; (**b**) y-direction. Appendix A for the x-direction and y-direction, respectively, have been attached for more detail.

**Figure 9 micromachines-15-00287-f009:**
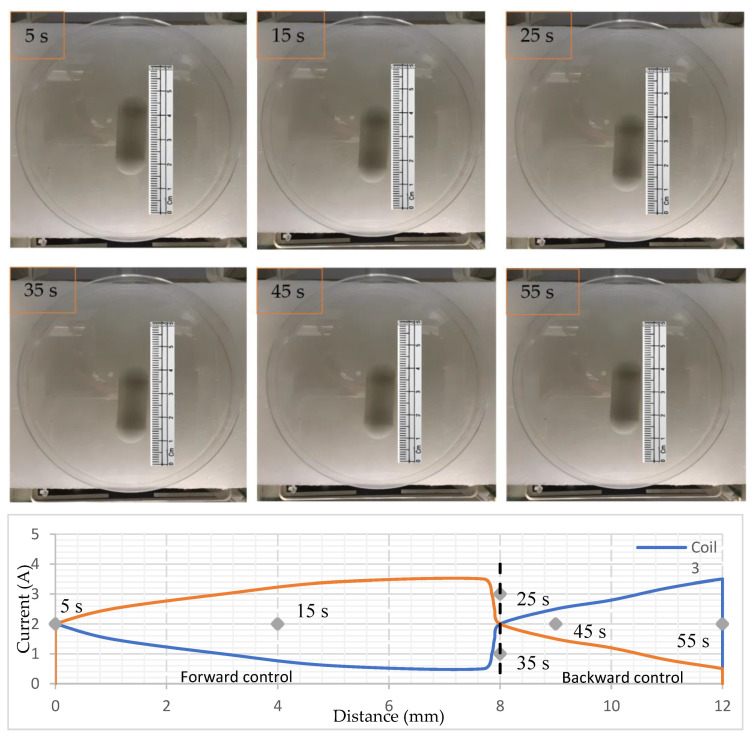
Forward and backward control locomotion and current in coils 3 and 4. Appendix A can be watched for more detail.

**Figure 10 micromachines-15-00287-f010:**
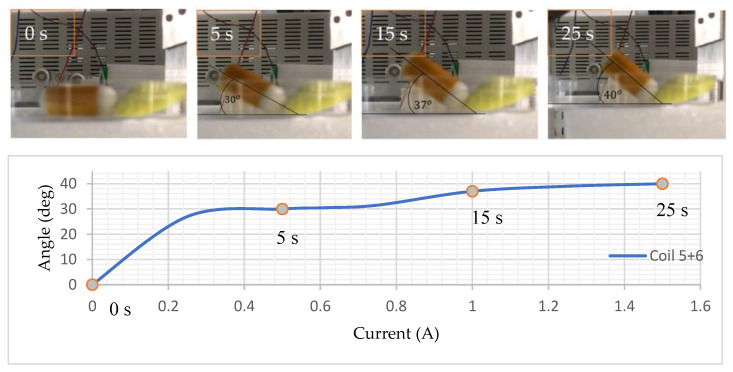
Rotational motion to point the robot with the lesion and the graphical explanation of changing angle with changing current. Appendix A has been attached for a detailed view.

**Figure 11 micromachines-15-00287-f011:**
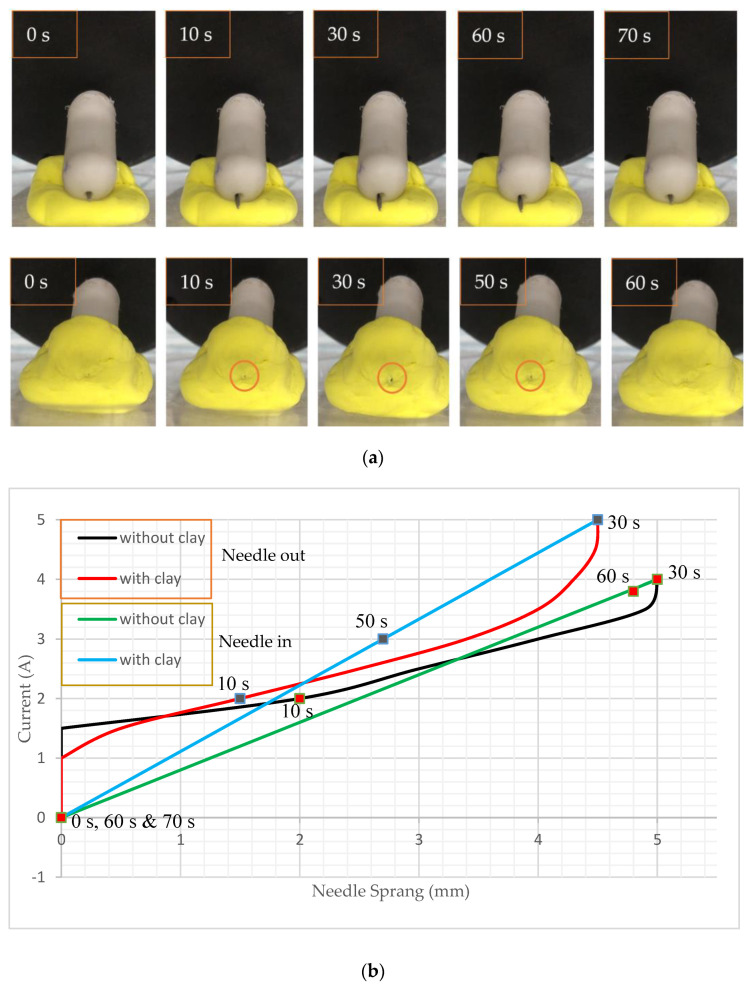
Functioning of biopsy mechanisms. (**a**) Procedures of tissue collection. (**b**) Comparison of input current with and without clay. Appendix A have been attached for detail.

**Table 1 micromachines-15-00287-t001:** Dimensions of the capsule robot.

Part’s Name	Number of Parts	Dimensions (mm)
Body	1	12 × 32.2
Permanent magnet 1	2	10-5 × 3
Permanent magnet 2	1	9.5-6 × 3.5
Spring	1	17
Needle	1	1.2 × 23
Camera module	1	
Transmitter module	1	

**Table 2 micromachines-15-00287-t002:** Parameters of the coils.

OCEC			MLCEC		
Parameter Name	Value	Unit	Parameter Name	Value	Unit
Outer length	~155	mm	Outer length	~75	mm
Inner length	130	mm	Inner length	10	mm
Wire diameter	0.85	mm	Wire diameter	0.87	mm
Number of turns	~1000	-	Number of turns	~1700	-
Coil resistance	14.3	Ω	Coil resistance	5.5	Ω
Distance from center	120	-			

**Table 3 micromachines-15-00287-t003:** Different input currents.

Operation Name	Minimum	Maximum	Operation Name	Minimum	Maximum
Translational motion in the x-direction	0.5 A (coil 2)	2.5 A (coil 1)	Rotational motion	0 A → 1.5 A	
Translational motion in the y-direction	0.5 A (coil 4)	2.5 A (coil 3)		(coil 5 + coil 6)	
Motion control forward & reverse	0.5 A → 3.5 A (coil 1)	3.5 A → 0.5 A (coil 2)	Biopsy function	0 A → 5 A (coil 7)	

## Data Availability

Data are contained within the article and Appendix A.

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
