# Peer review of "Magnetically Driven Biopsy Capsule Robot with Spring Mechanism"

_micromachines, 2024, doi:10.3390/mi15020287_

Round 1
Reviewer 1 Report
Comments and Suggestions for Authors
The paper introduces a magnetically controlled capsule robot equipped with a biopsy function designed for tissue extraction within the gastrointestinal (GI) tract. The external electromagnetic actuation system enables active locomotion and precise positioning of the robot, with the biopsy tool strategically positioned at the front side for improved defect area detection during movement. The experimental results demonstrate the robot's capability for translational locomotion, rotation, and biopsy functions. Nevertheless, certain issues require attention.
One specific concern is the need for a more detailed explanation of Figure 4. Additionally, the units for Figure 4a appear to be incomplete. Furthermore, it would be beneficial if the authors could undertake a comprehensive analysis of the magnetic field generated by the proposed magnetic field generator, incorporating parameters associated with this system.
To better adapt to the complex gastrointestinal environment, it is suggested that the authors conduct experiments to analyze the robot's climbing ability. Specifically, assessing the small-angle climbing ability on materials with a smoothness comparable to the gastrointestinal tract would provide valuable insights into the robot's performance in challenging scenarios.
Reviewer 2 Report
Comments and Suggestions for Authors
The paper “Magnetically Driven Biopsy Capsule Robot with Spring Mechanism” presents the development and initial validation of a ingenious, but not entirely new system for biopsy.
For a better understanding, some dimensions of the capsule should be indicated in figure 1.
Perhaps I have missed it, but please indicate specifically what is the targeted application? In this way the reader can understand better the environment and external forces acting upon the capsule.
What kind of tissue do you intend to punch? Have you recorded any considerable resistance during punching within the laboratory tests? The capsule tended to be pushed back, I presume.
Line colors in figure 11 (with or without clay) are very close, the contrast should be better.
A lot of magnetic capsules have been previously developed (see https://pubmed.ncbi.nlm.nih.gov/28437000/).
Please discuss them and emphasize your contribution in the discussion section of the paper.
Comments on the Quality of English LanguageThe English should also be improved, see paragraphs at lines 126, 136, 140, etc.
Round 2
Reviewer 2 Report
Comments and Suggestions for Authors
The authors have provided satisfactory responses to all my questions. The paper may be published